# Bone corticalization requires local SOCS3 activity and is promoted by androgen action via interleukin-6

Dae-Chul Cho[1,2], Holly J. Brennan[1,3], Rachelle W. Johnson[1,5], Ingrid J. Poulton[1], Jonathan H. Gooi[3], Brett A. Tonkin[1], Narelle E. McGregor[1], Emma C. Walker[1], David J. Handelsman[4], T.J. Martin[1,3] & Natalie A. Sims [1,3]

Long bone strength is determined by its outer shell (cortical bone), which forms by coalescence of thin trabeculae at the metaphysis (corticalization), but the factors that control this process are unknown. Here we show that SOCS3-dependent cytokine expression regulates bone corticalization. Young male and female *Dmp1Cre.Socs3*$^{f/f}$ mice, in which SOCS3 has been ablated in osteocytes, have high trabecular bone volume and poorly defined metaphyseal cortices. After puberty, male mice recover, but female corticalization is still impaired, leading to a lasting defect in bone strength. The phenotype depends on sex-steroid hormones: dihydrotestosterone treatment of gonadectomized female *Dmp1Cre.Socs3*$^{f/f}$ mice restores normal cortical morphology, whereas in males, estradiol treatment, or IL-6 deletion, recapitulates the female phenotype. This suggests that androgen action promotes metaphyseal corticalization, at least in part, via IL-6 signaling.

[1] St. Vincent's Institute of Medical Research, 9 Princes Street, Fitzroy, VIC 3065, Australia. [2] Department of Neurosurgery, Kyungpook National University Hospital, 130 Dongdukro, Jung-gu, Daegu 41944, Republic of Korea. [3] Department of Medicine at St. Vincent's Hospital, University of Melbourne, 41 Victoria Parade, Fitzroy, VIC 3065, Australia. [4] Department of Andrology, ANZAC Research Institute, University of Sydney, 3 Hospital Road, Concord, NSW 2139, Australia. [5]Present address: Division of Clinical Pharmacology, Vanderbilt University, 2215 Garland Avenue, 1255B MRB IV, Nashville, TN 37212, USA. Correspondence and requests for materials should be addressed to N.A.S. (email: nsims@svi.edu.au)

Cortical morphology at the metaphysis of long bones is a key determinant of bone strength. The metaphysis is the most common site of fragility fracture[1–3], and is the region in which cortical bone forms by coalescence of trabecular bone arising from the growth plate: termed "corticalization"[4]. This process is most active during growth, but also maintains cortical integrity in adulthood, when it continues at a slower rate[5]. Women with less bone in the metaphysis are more prone to fractures, and daughters of women with weak metaphyseal cortices also have poor corticalization[6], suggesting that the process of corticalization during growth determines adult fracture risk[5]. In addition, one of the reasons for increased prevalence of fractures in women is that their cortical bone is thinner than that of men[7], a sex difference that arises during the peri-pubertal period[8–10]. Despite its importance in determining bone strength, the mechanisms that control metaphyseal corticalization are not understood, in part because of the difficulties of obtaining normal cortical bone samples from growing children; this lack of knowledge means that treatments for osteoporosis, although successful at preventing vertebral fractures, have limited efficacy at non-vertebral sites[11, 12]. One of the key goals for new osteoporosis therapies is to promote bone formation, not only on trabecular surfaces, which contribute to vertebral strength, but also to promote bone formation on the periosteum (the outer cortical surface); this outcome would increase cortical thickness and strength, and prevent non-vertebral fractures.

Cortical thickness and periosteal growth are impaired in mice that lack IL-6[13, 14], and bone formation on the calvarial periosteum can be stimulated by any of several IL-6 family cytokines, including leukemia inhibitory factor (LIF)[15], cardiotrophin-1[16], and oncostatin M (OSM)[17]. This is mediated by a number of actions, including suppression of the Wnt inhibitor sclerostin in cells called osteocytes[17, 18], which form an interconnected cellular network that resides within both cortical and trabecular bone[19]. Targeted deletion of the common receptor subunit for these cytokines, gp130 (Il6st), in late osteoblasts (bone forming cells) and osteocytes resulted in poor cortical bone quality and increased bone width, suggesting an essential role for these cytokines in osteocytes to maintain cortical strength[20]. Furthermore, the ability of parathyroid hormone (PTH) to increase cortical width depended on gp130 expression in osteocytes[21]. This suggested that cortical bone might be strengthened by promoting gp130 signaling within the osteoblast–osteocyte network. We recently reported that murine OSM stimulates bone formation by promoting intracellular STAT3 signaling over STAT1[22], and amplification of the STAT3 signal downstream of gp130 restored bone mass in a mouse model of osteopenia[22]. This suggested that the beneficial effect of gp130 signaling on bone formation might be improved by specifically promoting STAT3 signaling[22].

A mechanism previously reported to promote intracellular STAT3 signaling is to delete SOCS3 (suppressor of cytokine signaling 3)[23–26], and indeed, Socs3 was the most strongly regulated target of murine OSM action in osteocyte-like cells[22]. SOCS3 is a ubiquitously expressed intracellular protein and provides negative feedback for STAT3 downstream of JAK/STAT-signaling cytokine receptors, including gp130, but also for leptin, G-CSF, and erythropoietin receptors[27]. We sought to determine whether targeting deletion of SOCS3 to osteocytes in vivo might stimulate bone formation. In doing so, we generated a mouse model of delayed metaphyseal corticalization. By studying this mouse, we find that the process of corticalization is controlled, not only by chondrocytes, but also by osteoblast-lineage cells, and that sex differences in corticalization may be explained, at least in part, by effects on local IL-6 signaling by androgens. This provides the first identification of a signaling pathway that controls corticalization.

## Results

**A unique high bone-mass phenotype that becomes sex divergent.** Dmp1Cre.Socs3^{f/f} mice were born at normal Mendelian ratios, and showed no gross abnormalities nor any change in body weight (Supplementary Fig. 1A). Significant knockdown of Socs3 mRNA was confirmed in flushed femora from both male and female 12-week-old Dmp1Cre.Socs3^{f/f} mice compared with Dmp1Cre controls (Fig. 1a); the reduction of the targeted gene mRNA in femoral samples to 50% is consistent with previous studies using this Dmp1Cre with ubiquitously expressed genes[20, 28, 29]; as previously noted the retained gene expression is likely due to non-osteoblast-lineage cells within the samples. As early as 2 weeks of age, Dmp1Cre.Socs3^{f/f} mice demonstrated a significantly greater trabecular bone volume (BV/TV) and trabecular number (Tb.N) than Dmp1Cre littermate controls (Fig. 1b, d, e). There was no significant alteration in longitudinal bone growth, indicated by femoral length (Fig. 1f), nor in periosteal growth, indicated by femoral periosteal circumference (Supplementary Fig. 1B), between control and Dmp1Cre.Socs3^{f/f} mice. At 6 weeks of age, BV/TV, Tb.N, and trabecular thickness (Tb.Th) continued to rise, and remained significantly higher in Dmp1Cre.Socs3^{f/f} mice than Dmp1Cre controls (Fig. 1b–d, g). This was particularly noticeable in female mice, which accrue less trabecular bone than males between 2 and 6 weeks of age, and therefore showed a proportionally greater BV/TV compared with the Dmp1Cre controls than males at 6 weeks of age. Representative thresholded images at 6 weeks (Fig. 1g) were consistent with this data, and the trabecular bone network lacked the normal level of organization. At 12 weeks of age, the bone phenotype showed a striking sex difference. In male mice, the high bone mass of Dmp1Cre.Socs3^{f/f} mice reversed, with BV/TV and Tb.N falling to levels below that of Dmp1Cre controls, and trabecular separation (Tb.Sp) becoming significantly higher; this low trabecular bone mass was retained until 26 weeks (Fig. 1b–i). In contrast, female Dmp1Cre.Socs3^{f/f} mice continued to accrue trabecular bone: Tb.Th, Tb.N, and BV/TV all continued to rise, with BV/TV reaching levels sevenfold higher than that of controls at 12 weeks of age (Fig. 1b–h). Representative thresholded images at 12 weeks showed a distinct high bone-mass phenotype in Dmp1Cre.Socs3^{f/f} mice; the trabecular bone was chaotic in its organization and mainly distributed around a central empty core (Fig. 1h). The phenotype gradually recovered, with BV/TV reduced, but remaining high at 16 weeks of age, and then meeting normal values, with only high Tb.Sp and Tb.Th at 26 weeks of age (Fig. 1b–i). At 26 weeks of age, representative images from female mice showed a paucity of trabeculae, and cortical bone appeared somewhat porous compared to controls (Fig. 1i).

High trabecular bone mass in growing mice is caused either by excessive bone formation (osteosclerosis)[30] or insufficient bone resorption (osteopetrosis)[31]. Histomorphometry on 6-week-old tibiae revealed that the high trabecular bone mass of both male and female Dmp1Cre.Socs3^{f/f} mice was explained by a greater level of bone formation, with a greater than twofold elevation in osteoblast and osteoid surfaces (Fig. 2a, b), as well as abundant calcein labeling on trabecular surfaces (Fig. 2d). In normal lamellar trabecular bone, calcein labels are detected as two distinct lines (Fig. 2e, left panel), but in both female and male mice, a high proportion of calcein labels in the secondary spongiosa were diffuse and non-linear, indicative of woven-bone formation (Fig. 2e). The high level of bone formation was not matched with a high level of resorption: osteoclast numbers were not significantly increased (Fig. 2c). There were no indicators of osteopetrosis, such as cartilage remnants within the trabecular bone of the secondary spongiosa, nor any extension of the growth plate hypertrophic zone (Fig. 2f), confirming that Dmp1Cre.Socs3^{f/f} mice are osteosclerotic at 6 weeks of age.

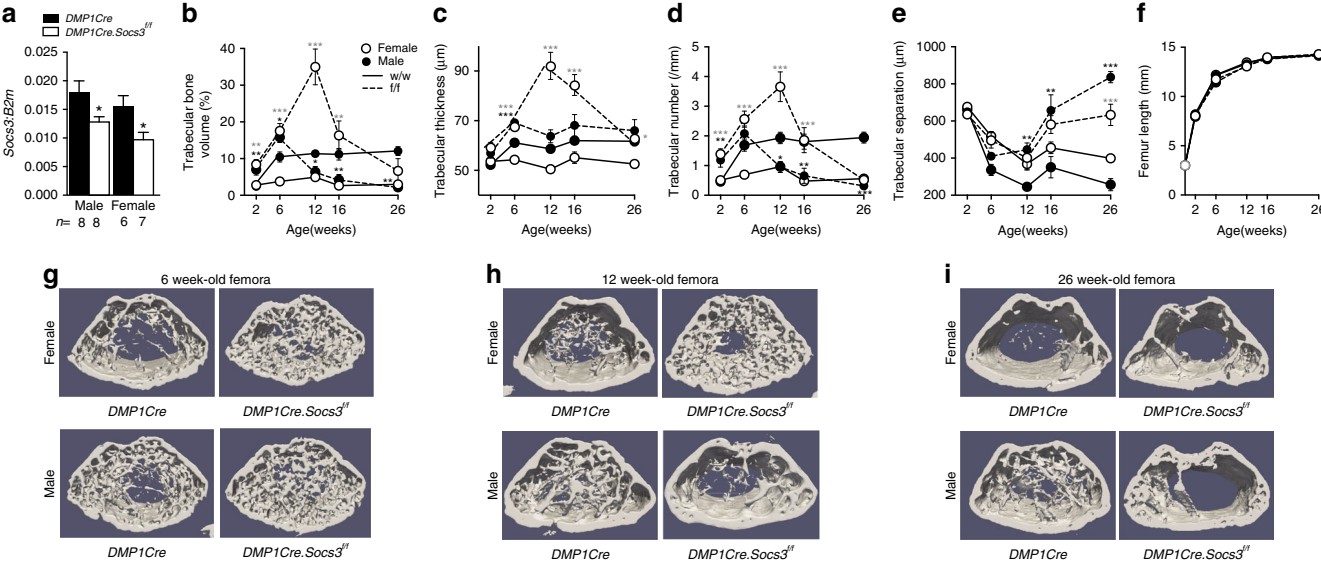

**Fig. 1** Sex divergence in the adult bone phenotype in male and female *Dmp1Cre.Socs3^{f/f}* mice vs *Dmp1Cre* controls. **a** mRNA levels of *Socs3* in femora, flushed of marrow, from 12-week-old *Dmp1Cre.Socs3^{f/f}* and *Dmp1Cre* littermates; values are mean + SEM, *n* shown below graph, \**p* < 0.05 vs *Dmp1Cre* by two-way ANOVA with Sidak post-hoc test. Femoral trabecular bone volume (**b**), trabecular thickness (**c**), trabecular number (**d**), trabecular separation (**e**), and femoral length (**f**) of 2, 6, 12, 16, and 26-week-old male (*filled circles*) and female (*open circles*) *Dmp1Cre.Socs3^{f/f}* (*dashed lines*—f/f) and *Dmp1Cre* (*solid lines*—w/w) mice. Values are mean ± SEM; numbers are as follows in the order female w/w, female f/f, male w/w, male f/f: 2 weeks (7, 8, 8, 7); 6 weeks (9, 11, 12, 10); 12 weeks (11, 10, 10, 9); 16 weeks (9, 8, 11, 7); 26 weeks (9, 9, 9, 10). \**p* < 0.05; \*\**p* < 0.01; \*\*\**p* < 0.001; vs *Dmp1Cre* of the same age and sex by two-way ANOVA with Sidak post-hoc test; gray stars indicate significant differences in females. Representative thresholded micro-CT images of the metaphyseal region used for analysis in 6-week-old (**f**), 12-week-old (**g**), and 26-week-old (**h**) mice showing time-dependent and sex-dependent modifications in trabecular structure. Images are not to scale

At 12 weeks of age, a sex divergence in bone cell activities emerged. Although both male and female *Dmp1Cre.Socs3^{f/f}* mice still demonstrated osteoblast and osteoid surfaces approximately double that of controls (Fig. 3a, b), male *Dmp1Cre.Socs3^{f/f}* mice showed a significantly higher osteoclast surface than their controls (Fig. 3c). This was co-incident with the reduction in BV/TV and Tb.N in male *Dmp1Cre.Socs3^{f/f}* mice (Fig. 1b, d). The high BV/TV in female mice still was not associated with any markers of osteopetrosis (Fig. 3d), but showed extensive woven-bone formation, particularly near the endocortical regions (Fig. 3e, f). By 26 weeks of age, both male and female *Dmp1Cre.Socs3^{f/f}* mice showed high levels of both bone formation and resorption, suggested by their high Tb.Sp., and confirmed by serum markers (histomorphometry at this age is difficult due to the paucity of trabecular bone in female mice) (Fig. 3g, h).

**Delayed corticalization in female mice impairs bone strength**. Micro-CT imaging without thresholding at 26 weeks of age showed less consolidation of cortical bone, even in the femoral diaphysis of both male and female *Dmp1Cre.Socs3^{f/f}* mice (see asterisks in Fig. 4a and b), suggesting impaired strength. Although cortical area and the ultimate force reached before bones broke under 3 point-bending conditions were not significantly different between the two genotypes (Fig. 4c, d), female *Dmp1Cre.Socs3^{f/f}* mice, but not males, showed significantly lower material strength of their cortical bone, as indicated by lower work to failure (Fig. 4e) and post-yield deformation (Fig. 4f, g). This suggested that the process of cortical bone formation was impaired in female mice. Indeed, when generating the data in Fig. 1, although trabecular bone mass was greater in female *Dmp1Cre.Socs3^{f/f}* mice it was very difficult to distinguish between trabecular and cortical bone, suggesting defective corticalization led to impaired strength even after the trabecular phenotype had recovered, in older mice. In males, which showed reduced

trabecular bone mass at this age, although corticalization is improved, the cortical bone appears thicker than normal and is highly porous (Figs. 1g–i, 4b).

**Androgens promote, and estradiol inhibits, corticalization**. The sex-steroid dependency of the corticalization defect in *Dmp1Cre.Socs3^{f/f}* mice was investigated according to the protocol in Fig. 5a. Vertebrae of female 12-week-old *Dmp1Cre.Socs3^{f/f}* mice did not have high BV/TV as their femora did, but demonstrated significantly lower BV/TV than controls (Fig. 5b–d). This provides further support for the concept that delayed corticalization, a process required to form the thickened cortical bone of the limbs, but not for the thin cortical bone in the vertebrae, may drive the high femoral BV/TV of female *Dmp1Cre.Socs3^{f/f}* mice. As observed in the long bones, male *Dmp1Cre.Socs3^{f/f}* mice also showed low BV/TV compared to controls (Fig. 5e–g). Since the marked increase in femoral BV/TV might mask the effects of sex-steroid implants, vertebral scans were used to confirm their effectiveness in protecting bone from gonadectomy-induced loss. In both *Dmp1Cre* and *Dmp1Cre.Socs3^{f/f}* mice, ovariectomy (OVX) and orchiectomy (ORX) reduced BV/TV, and provision of silastic implants of estradiol ($E_2$) and non-aromatizable dihydrotestosterone (DHT) prevented the vertebral trabecular bone loss associated with gonadectomy without causing the significant sclerosis that is associated with high sex-steroid doses[32] (Fig. 5b–g). This confirmed that the doses used are effective replacement doses for bone loss following gonadectomy in male and female mice.

In female *Dmp1Cre* (control) mice, OVX reduced trabecular bone mass, and, as in vertebral bone, this was protected by both $E_2$ and DHT (Fig. 6a). In *Dmp1Cre.Socs3^{f/f}* mice, raw micro-CT images again showed the difficulty of distinguishing between trabecular and cortical bone (Fig. 6b). OVX improved corticalization in *Dmp1Cre.Socs3^{f/f}* female mice, although the cortical bone

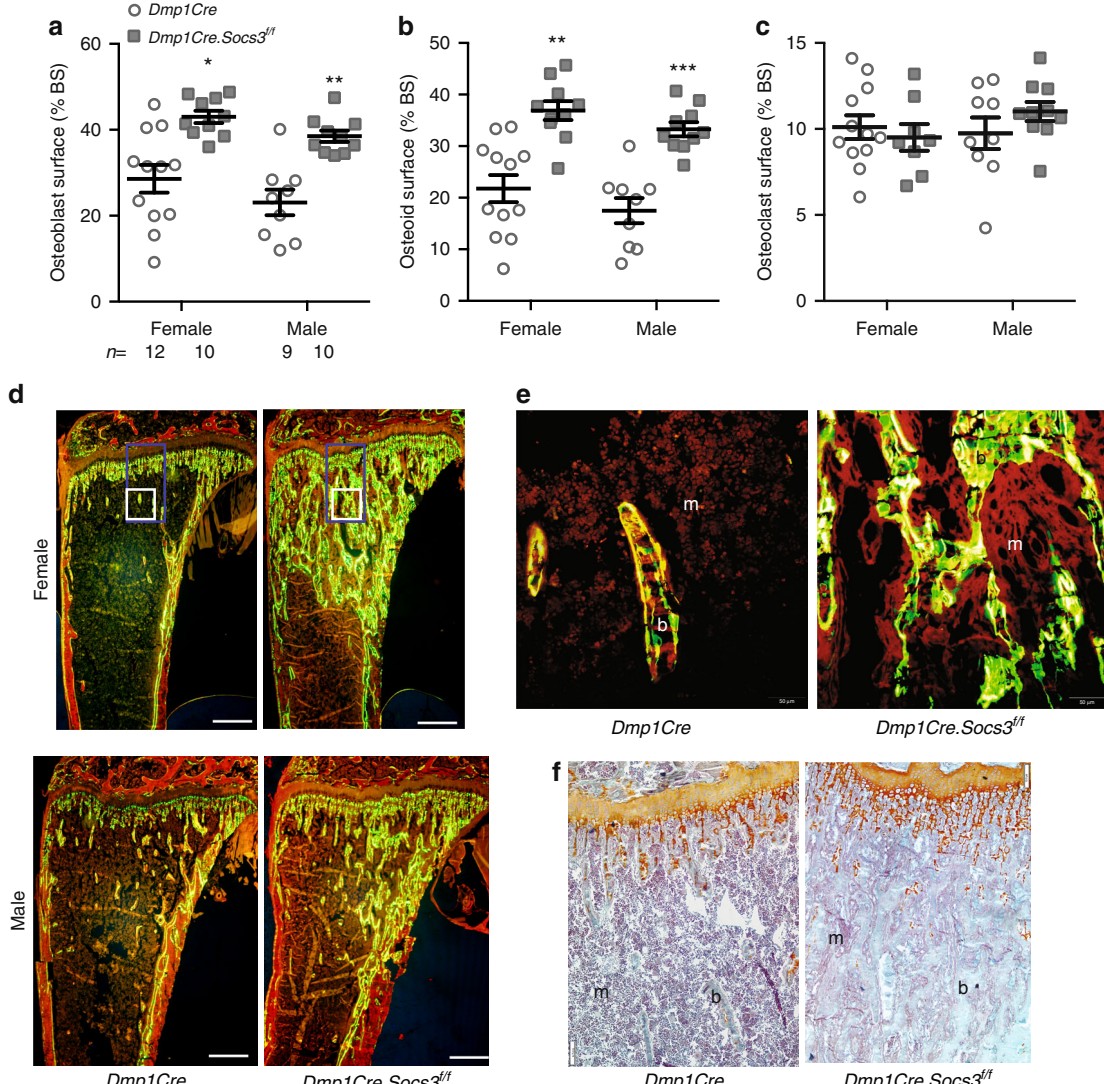

**Fig. 2** High bone formation in 6-week-old male and female *Dmp1Cre.Socs3^{f/f}* mice. **a–c** Histomorphometric analysis of osteoblast surface (**a**), osteoid surface (**b**) and osteoclast surface (**c**) in the secondary spongiosa of 6-week-old male and female *Dmp1Cre.Socs3^{f/f}* (*gray squares*) and *Dmp1Cre* (*open circles*) tibiae; values are mean ± SEM, *n* is shown below (**a**). *\*p < 0.05; \*\*p < 0.01; \*\*\*p < 0.001; vs Dmp1Cre* of the same sex by two-way ANOVA with Sidak post-hoc test. **d** Representative images of calcein labeling of bone (b) in male and female *Dmp1Cre.Socs3^{f/f}* and *Dmp1Cre* mice; m, marrow. Scale bar = 500 μm. **e** High-power images of the boxed regions in panel (**b**) showing lamellar calcein labels in the secondary spongiosa in female *Dmp1Cre* tibiae and woven-bone formation in the same region in female *Dmp1Cre.Socs3^{f/f}* mice. Scale bar = 50 μm. **e** Safranin O staining shows no elevation in cartilage remnants (*orange*) within the trabecular bone (b) of female *Dmp1Cre.Socs3^{f/f}* mice compared with *Dmp1Cre* mice; scale bar = 50 μm

still appeared porous; this OVX effect was prevented by E₂ and reproduced by DHT treatment (Fig. 6b). ORX reduced trabecular bone mass in male *Dmp1Cre* and *Dmp1Cre.Socs3^{f/f}* mice, and this was prevented by DHT (Fig. 6c, d). E₂ treatment of male *Dmp1Cre.Socs3^{f/f}* mice recapitulated the chaotic organisation of bone characteristic of the female *Dmp1Cre.Socs3^{f/f}* phenotype, indicating that estradiol prevents corticalization via SOCS3-dependent mechanisms.

"Thickened" cortical bone was measured throughout the metaphyseal region and partitioned into the spongy, incompletely coalesced cortical component vs normal compact cortical bone. In *Dmp1Cre* control mice, there was no incompletely formed (spongy) cortical bone (Fig. 6e). There was a significantly greater cortical thickness in female *Dmp1Cre.Socs3^{f/f}* mice compared to controls, but approximately one-fourth of it was incompletely formed spongy bone (Fig. 6e). DHT rescued this phenotype, allowing complete consolidation of cortical bone to occur, no spongy bone was detected, but cortical thickness still remained

higher in *Dmp1Cre.Socs3^{f/f}* mice than both sham-operated and DHT-treated controls (Fig. 6e). Male *Dmp1Cre.Socs3^{f/f}* mice had thickened cortical bone compared to *Dmp1Cre*, but it did not contain spongy bone; however, estradiol treatment of male *Dmp1Cre.Socs3^{f/f}* mice resulted in a high proportion of spongy bone within the cortex, completely recapitulating the female *Dmp1Cre.Socs3^{f/f}* phenotype (Fig. 6f).

This data was confirmed using threshold-based measurements of cortical porosity in the metaphysis, the region in which cortical bone is least developed. Male and female *Dmp1Cre* control mice exhibited minimal cortical porosity (Fig. 6g, h), whereas female *Dmp1Cre.Socs3^{f/f}* mice had highly porous cortical bone (Fig. 6g). DHT treatment prevented this high level of cortical porosity, lowering cortical porosity levels so that they were no longer significantly different from controls (Fig. 6g). Estradiol treatment maintained the high level of cortical porosity (Fig. 6g). In male *Dmp1Cre.Socs3^{f/f}* mice, estradiol prevented trabecular coalescence, resulting in a very high cortical porosity, similar to that of

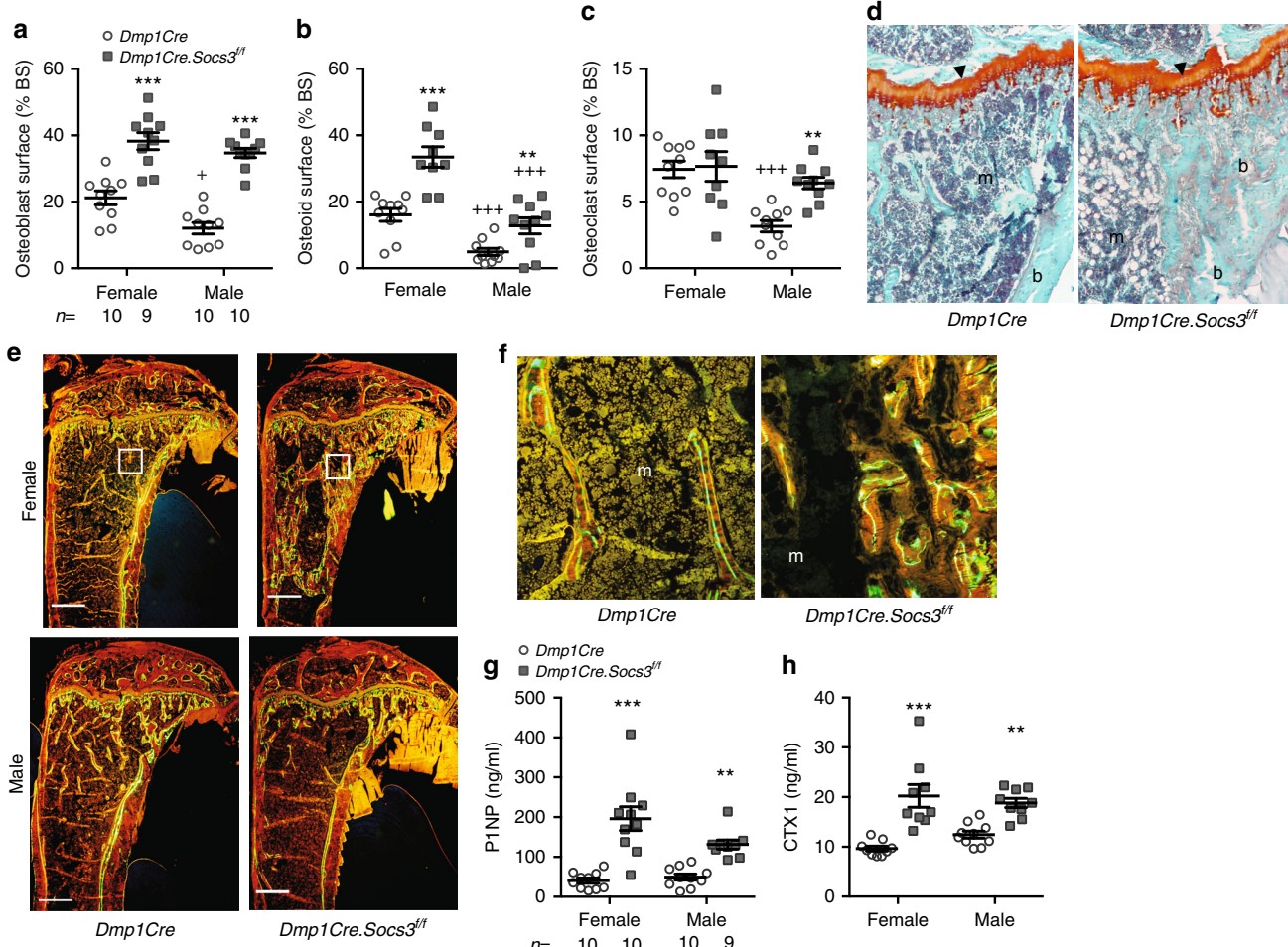

**Fig. 3** Sex divergence of the *Dmp1Cre.Socs3^{f/f}* phenotype explained by a male-specific effect on osteoclast numbers at 12 weeks of age. **a–c** Histomorphometric analysis of osteoblast surface (**a**), osteoid surface (**b**), and osteoclast surface (**c**) in the secondary spongiosa of 12-week-old male and female *Dmp1Cre* (*open circles*) and *Dmp1Cre.Socs3^{f/f}* (*gray squares*) tibiae; values are mean ± SEM, $n$ is shown for each group below (**a**). \*\*$p < 0.01$; \*\*\*$p < 0.001$ vs *Dmp1Cre* of the same sex; $^{+}p < 0.05$, $^{+++}p < 0.001$ vs female of the same genotype by two-way ANOVA with Sidak post-hoc test. **d** Safranin O staining shows no retention of cartilage remnants (orange) within the trabecular bone (b) of female *Dmp1Cre.Socs3^{f/f}* mice compared with *Dmp1Cre* mice; m marrow; scale bar = 50 μm. **e** Representative images of calcein labeling of bone in male and female 12-week-old *Dmp1Cre.Socs3^{f/f}* and *Dmp1Cre* mice. Scale bar = 500 μm. **f** High-power images from the boxed regisons shown in **e** showing lamellar bone formation in the secondary spongiosa in female *Dmp1Cre* mice (left), but woven-bone formation in the same region in female *Dmp1Cre.Socs3^{f/f}* mice (right); scale bar = 50 μm. **g, h** Biochemical markers of bone formation (P1NP) and resorption (CTX-1) in 26-week-old male and female *Dmp1Cre.Socs3^{f/f}* (*open circles*) and *Dmp1Cre* (*gray squares*) mice; values are mean ± SEM, $n$ is shown for each group below (**g**). \*$p < 0.05$; \*\*$p < 0.01$; \*\*\*$p < 0.001$; vs *Dmp1Cre* of the same sex by two-way ANOVA with Sidak post-hoc test

female *Dmp1Cre.Socs3^{f/f}* mice (compare Fig. 6h and g). In more mature bone, closer to the diaphysis, in the "lower metaphysis", cortical porosity was also higher in both male and female *Dmp1Cre.Socs3^{f/f}* mice compared with sex-matched controls (Fig. 6i, j). As the cortical bone in this central region would have already been established prior to 6 weeks of age, when hormonal intervention commenced, DHT was unable to rescue this phenotype, but surprisingly, estradiol treatment in male *Dmp1Cre.Socs3^{f/f}* mice still resulted in a higher level of cortical porosity in this region. This suggests that the process of corticalization continues for some time after the cortical shell is established. Calcein labeling at a 6 day interval before tissue collection (Fig. 6k, l) showed that DHT treatment rescued the chaotic bone formation in female *Dmp1Cre.Socs3^{f/f}* mice and improved it in males, restoring the normal smooth appearance of the endocortical surface. In contrast $E_2$ resulted in a high level of woven-bone formation in both male and female *Dmp1Cre.Socs3^{f/f}* mice (Fig. 6k, l).

**Androgens promote corticalization via IL-6.** As pathologies caused by macrophage, liver and T cell-specific knockouts of SOCS3 have been attributed to hyperactive IL-6 by virtue of their rescue when IL-6 was deleted[23, 26, 33], we crossed *Dmp1Cre. Socs3^{f/f}* mice with IL-6 null mice (Fig. 7a). Deletion of IL-6 did not rescue the defective corticalization in either male or female *Dmp1Cre.Socs3^{f/f}* mice. However, deletion of IL-6 ameliorated the sex-specific difference in the *Dmp1Cre.Socs3^{f/f}* phenotype such that femoral cortical porosity of male *Dmp1Cre.Socs3^{f/f}.IL-6^{-/-}* was no longer significantly lower than that of female *Dmp1Cre. Socs3^{f/f}* mice (Fig. 7b). Calcein labeling in the cortex of the tibial upper secondary spongiosa was more abundant at the inner endocortical surface and extended fully through to the outer periosteal edge, indicating that trabecular bone was not becoming consolidated into a thickened cortex with a smooth endocortical surface as in control mice of this age (Fig. 7c). Von Kossa staining confirmed an impairment in corticalization in *Dmp1Cre.Socs3^{f/f}. IL-6^{-/-}* male mice, but not to the same extent as in female

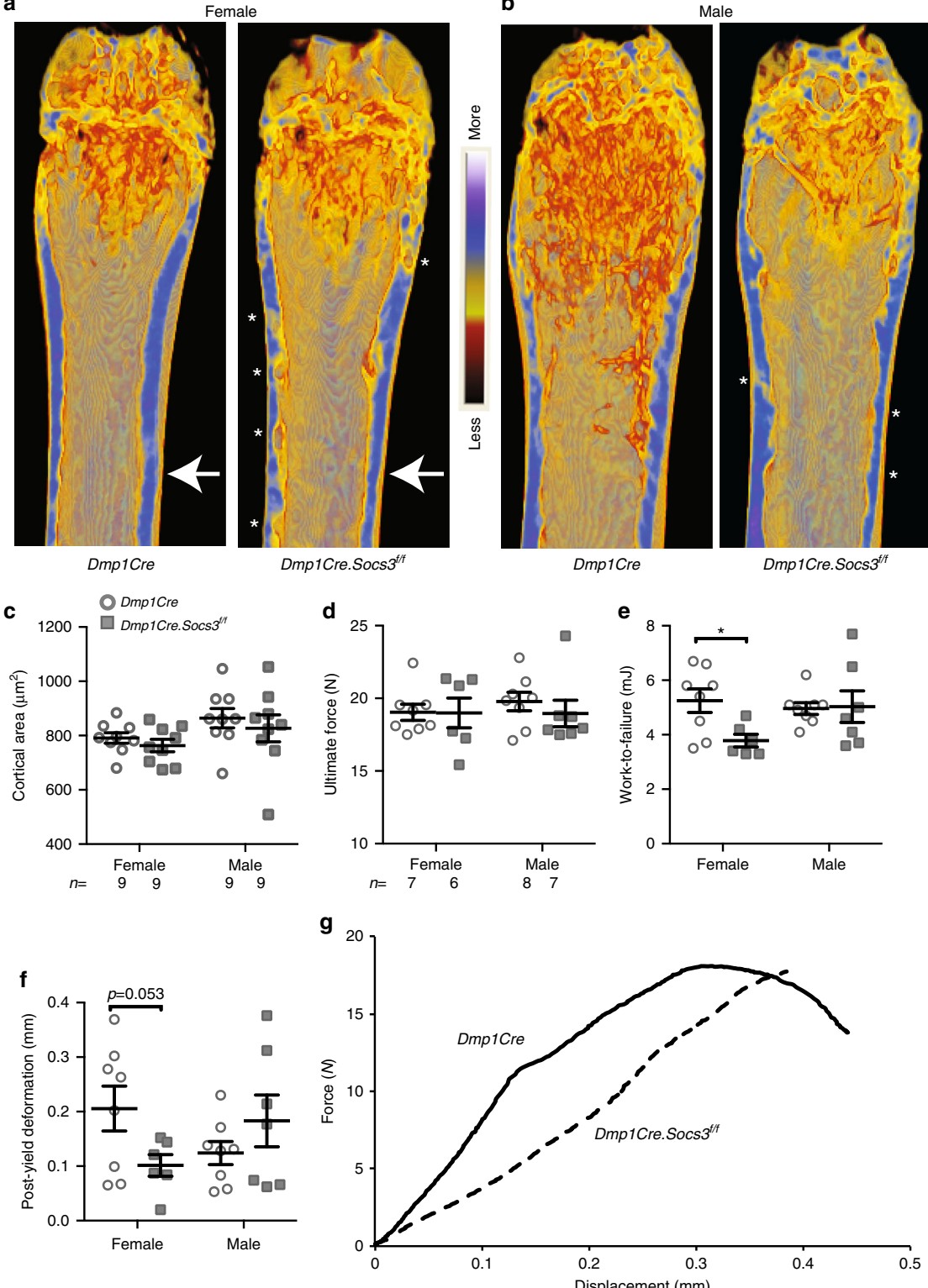

**Fig. 4** Sex-specific delay in corticalization in female *Dmp1Cre.Socs3f/f* mice leads to reduced bone strength at 26 weeks of age. **a**, **b** Representative coronal cross sections of female (**a**) and male (**b**) femora showing levels of mineralization (scale at the center of the images shows *red* = less mineralized, while *blue* = more mineralized); arrows indicate approximate location of load during 3 point-bending; *asterisks* alongside cortical bone denote regions of poor cortical consolidation. **c** Cortical bone area in 26-week-old male and female *Dmp1Cre* (*open circles*) and *Dmp1Cre.Socs3f/f* (*gray squares*) femora; values are mean ± SEM, *n* is shown below the graph. **d**–**f** Results of 3 point-bending tests of femora from 26-week-old male and female *Dmp1Cre* (*open circles*) and *Dmp1Cre.Socs3f/f* (*gray squares*) femora; values are mean ± SEM, *n* is shown below graph (**d**). *$p < 0.05$; **$p < 0.01$ vs *Dmp1Cre* of the same sex by two-way ANOVA with Sidak post-hoc test. **g** Representative load-displacement curves of female *Dmp1Cre* (*solid line*) and *Dmp1Cre.Socs3f/f* (*dashed line*) femora

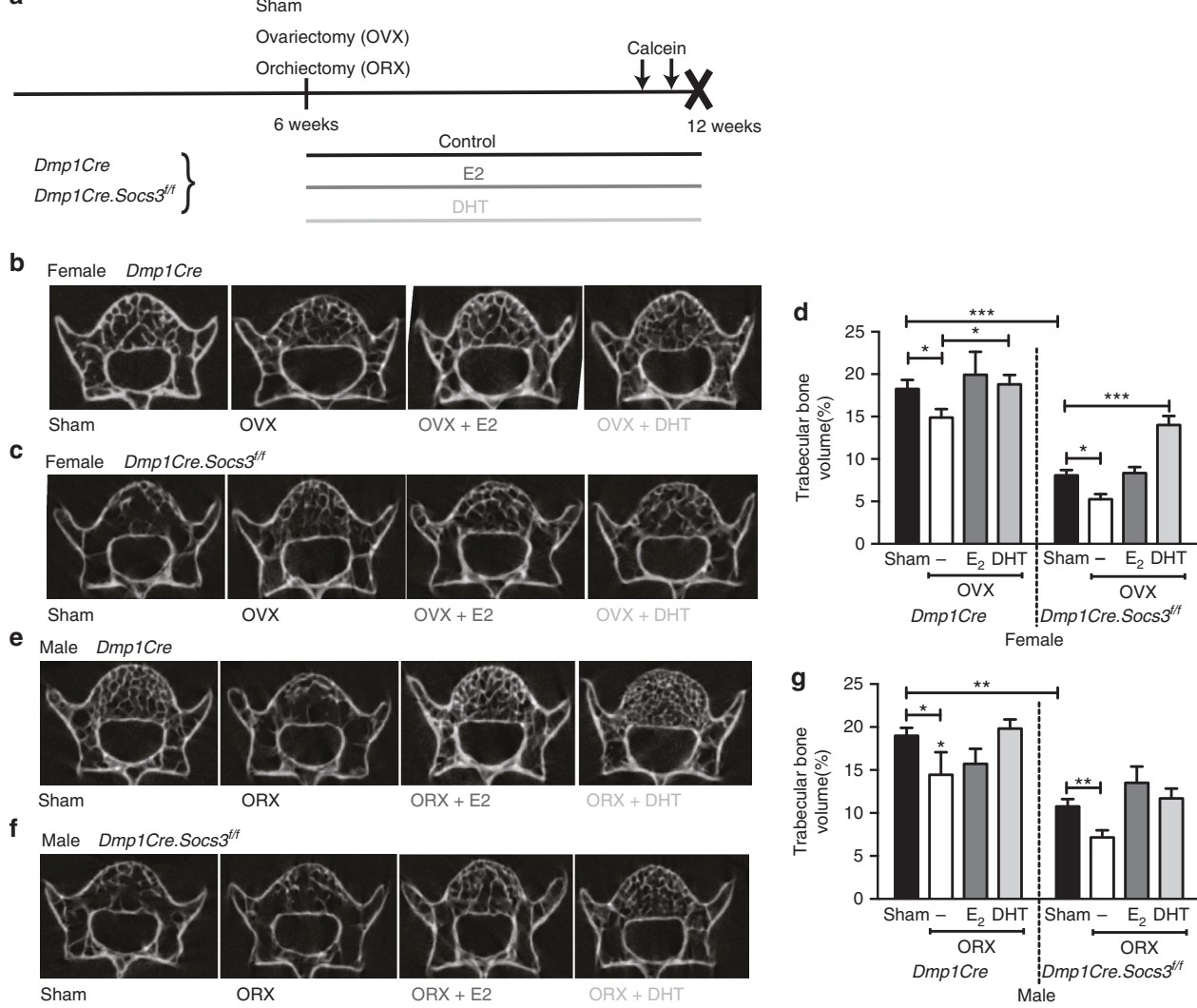

**Fig. 5** Sex hormone treatment protects against gonadectomy-induced trabecular bone loss in the 5th lumbar vertebrae of female and male *Dmp1Cre.Socs3^{f/f}* and *Dmp1Cre* mice. **a** Schematic showing experimental protocol including sham operation, ovariectomy (OVX), or orchiectomy (ORX) at 6 weeks of age, and treatment with silastic implants of dihydrotestosterone (DHT) or estradiol (E2) from 6–12 weeks of age. **b**–**g** Micro-CT analysis including representative transaxial images of the 5th lumbar vertebrae of female *Dmp1Cre* (**b**), *Dmp1Cre.Socs3^{f/f}* (**c**), and male *Dmp1Cre* (**e**), *Dmp1Cre.Socs3^{f/f}* (**f**) mice from each treatment group and vertebral trabecular bone volume of female (**d**) and male (**g**) *Dmp1Cre* and *Dmp1Cre.Socs3^{f/f}* mice subjected to gonadectomy and steroid treatments as outlined in **a**. Values are mean + SEM, $n = 6$ per group. *$p < 0.05$; **$p < 0.01$; ***, $p<0.001$ for comparisons indicated in the figure

*Dmp1Cre.Socs3^{f/f}* mice (Fig. 7d). This suggests that the androgen-dependent rescue of delayed corticalization in male *Dmp1Cre. Socs3^{f/f}* mice is caused, at least in part, by an IL-6-dependent mechanism.

## Discussion

We describe a unique mouse model indicating that metaphyseal corticalization is delayed by promoting the intracellular signaling of SOCS3-dependent cytokines (Fig. 7e) within *Dmp1Cre*-expressing cells (osteoblasts and osteocytes). The recovery from this phenotype is sex-dependent, with the delay in corticalization being longer in female mice. This extended delay in female corticalization was prevented by treatment with the pure androgen DHT, which is non-aromatizable, and was fully recapitulated in males by estradiol treatment, and partially recapitulated by IL-6 knockout, suggesting that pure androgen action (mainly mediated by testosterone in vivo) promotes corticalization through an IL-6-dependent pathway (Fig. 7e). This is the first identification of a

mechanism by which corticalization occurs; targeting the causative cytokine may provide new therapeutic approaches to strengthen cortical bone, particularly in the context of fracture healing.

The defect in metaphyseal corticalization of *Dmp1Cre.Socs3^{f/f}* mice appears to be a unique morphological phenotype, suggesting a unique role in this process for cytokines inhibited by SOCS3. The distinct aspects of the phenotype of 12-week-old female *Dmp1Cre.Socs3^{f/f}* mice include (1) the specific location of high trabecular bone mass in the metaphyseal endocortical region of long bones, (2) the disorganized morphology of the high trabecular bone mass, (3) the hollow region lacking trabecular bone in the center of the femoral metaphysis, (4) the delay in cortical bone formation, and (5) the contrasting low trabecular bone mass in vertebral bodies. Other published mouse models that exhibit increased bone formation without increased osteoclastogenesis, i.e., osteosclerosis, have robust trabecular bone throughout the metaphysis, robust metaphyseal cortices, trabecular bone that fills the marrow space (rather than leaving a central core with low

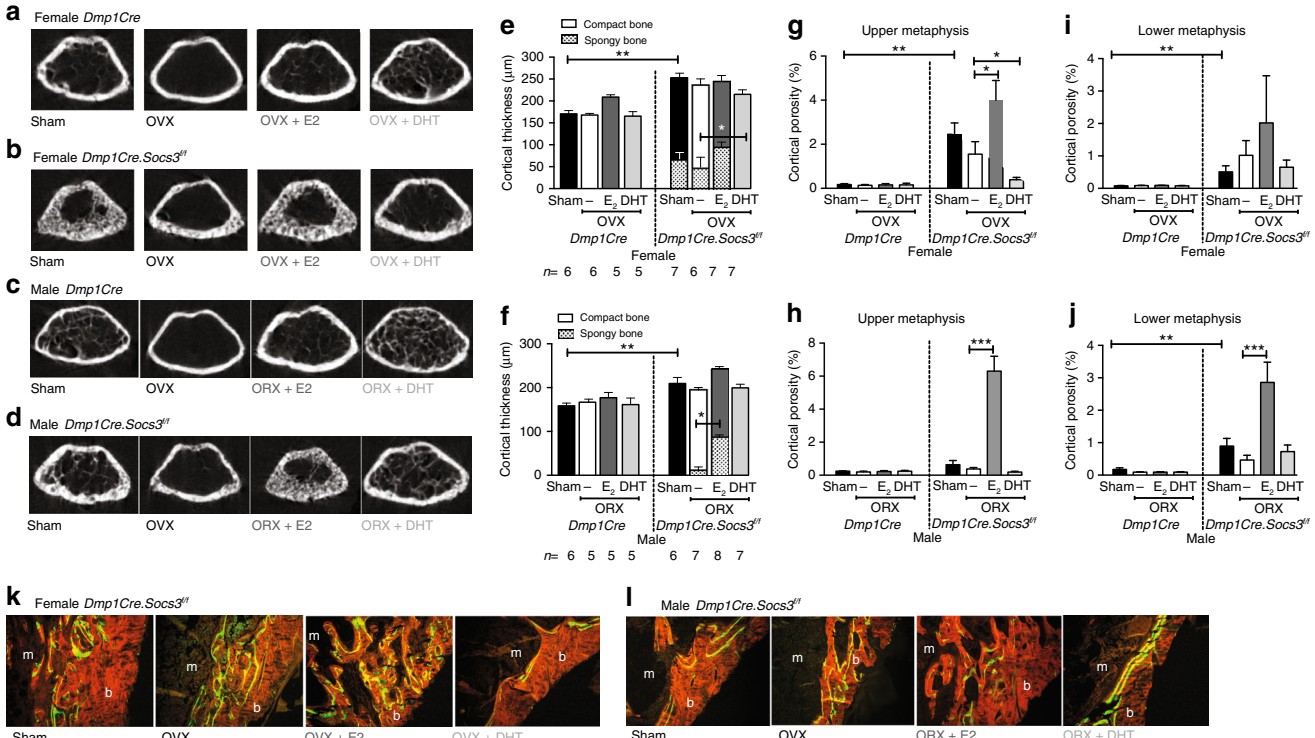

**Fig. 6** Androgen treatment rescues the female *Dmp1Cre.Socs3^{f/f}* defect in corticalization, whereas estradiol treatment of male *Dmp1Cre.Socs3^{f/f}* mice recapitulates the female phenotype. Treatment protocol is shown in Fig. 5a. **a, b, e, g, i, k** Female mice. **c, d, f, h, j** Male mice. Shown are transaxial cross sections at the top of the femoral metaphyseal region of analysis in female (**a, b**) and male (**c, d**) *Dmp1Cre* (**a, c**), and *Dmp1Cre.Socs3^{f/f}* (**b, d**) mice sham-operated or gonadectomized (ovariectomy OVX, orchiectomy ORX) and treated with silastic implants of dihydrotestosterone (DHT) or estradiol (E2) from 6–12 weeks of age. Data shown are cortical thickness, including both compact cortical bone, and porous (spongy) bone, the latter shown in hatched bars (**e, f**), and cortical porosity measured at both the upper (**g, h**) and lower (**i, j**) metaphysis in female (**e, g, i**) and male (**f, h, j**) *Dmp1Cre* and *Dmp1Cre.Socs3^{f/f}* mice sham-operated or gonadectomized and treated with silastic implants of DHT or E2 from 6–12 weeks of age. Values are mean + SEM, *n* is shown for each group in **e, f**; *$p < 0.05$; **$p < 0.01$ for comparisons indicated in the figure by two-way ANOVA with Sidak post-hoc test. **k, l** Representative histological images of calcein labels in the corticalization zone on the lateral side of the tibial metaphysis for each treatment group. Orange = bone; green = calcein; m = marrow; b = bone

trabecular bone mass), increased cortical strength (where it has been tested), and increased trabecular bone mass in both femora and vertebrae. These include both mice with global deletion or overexpression, such as sclerostin null mice[34], Dkk1 null mice[35], ΔFosB-overexpressing mice and Fra1-overexpressing mice[30, 36], Neuropeptide Y2 null mice[37], and mice with primary modifications directed towards osteoblasts and osteocytes, such as *Dmp1Cre*-directed *Lrp5* high bone-mass mutants[38], *Dmp1Cre*-directed *Mef2c* deletion[39], and both Osteocalcin-Cre-directed and *Dmp1Cre*-directed *Lrp4* deletion[40]. The morphological defect in the adult female *Dmp1Cre.Socs3^{f/f}* mouse is also different to osteopetrotic phenotypes (states of impaired bone resorption). Although osteopetrotic mice exhibit high trabecular mass and low cortical thickness, their phenotype is characterized by impaired resorption or osteoclastogenesis with a widened hypertrophic zone of the growth plate and increased cartilage remnants within trabecular bone that fills the metaphyseal region. Examples of osteopetrotic phenotypes that exhibit these features include mice with global deletion of *c-src*[41], *Pyk2*[31], cardiotrophin-1[16], IL-11 receptor[13], M-CSF[42] or RANKL[43], and in mice with *Dmp1Cre*-directed deletion of RANKL[44, 45]. The phenotype we observe is dissimilar to all of these previously described mice, and suggests a unique function for SOCS3-dependent cytokines that is required for the process of metaphyseal corticalization.

Receptors for, and responses to, numerous cytokines that require SOCS3 for negative feedback have been reported in osteoblasts and osteocytes, indicating that the delay in corticalization could be explained by hyperactivity of a range of possible cytokines. These include G-CSF[46], leptin[47], gp130 (the common receptor subunit used by all IL-6 family cytokines)[48], as well as IL-6 family cytokine-specific receptors such as IL-6R[49], IL-11R[48], LIFR[50], OSMR[17], and the CNTFR[51], which is utilized by a subset of IL-6 family cytokines[52]. Similarly, ligands for each of these receptors are produced within the local environment, including by osteoblasts and osteocytes[16, 48, 53–55]. A role for these cytokines in determining cortical bone development is also supported by reports of thin cortices in mice null for IL-6[13, 14], IL-11R[13], or CNTF[51] and greater cortical thickness in leptin receptor[56] and OSMR null mice[17], although these are not cell-specific knockouts and only in OSMR null mice were measurements made specifically at the metaphysis. The region-specific phenotype of the *Dmp1Cre.Socs3^{f/f}* mouse suggests that local suppression by SOCS3 of the intracellular action of one or more of these cytokines is required for metaphyseal corticalization.

That corticalization was delayed until 6 weeks of age in both males and females indicates that SOCS3 is required for corticalization during bone growth in both sexes. Sex divergence after that age indicates that sex hormones also influence metaphyseal corticalization through SOCS3. Differences in skeletal structure, particularly in the cortical thickness of long bones are well described in male and female mammals[8, 10]. In humans, this sex difference arises because boys have a longer period of growth, during which the periosteum continues to expand[8, 10], whereas girls form bone on the endocortical metaphysis in a process

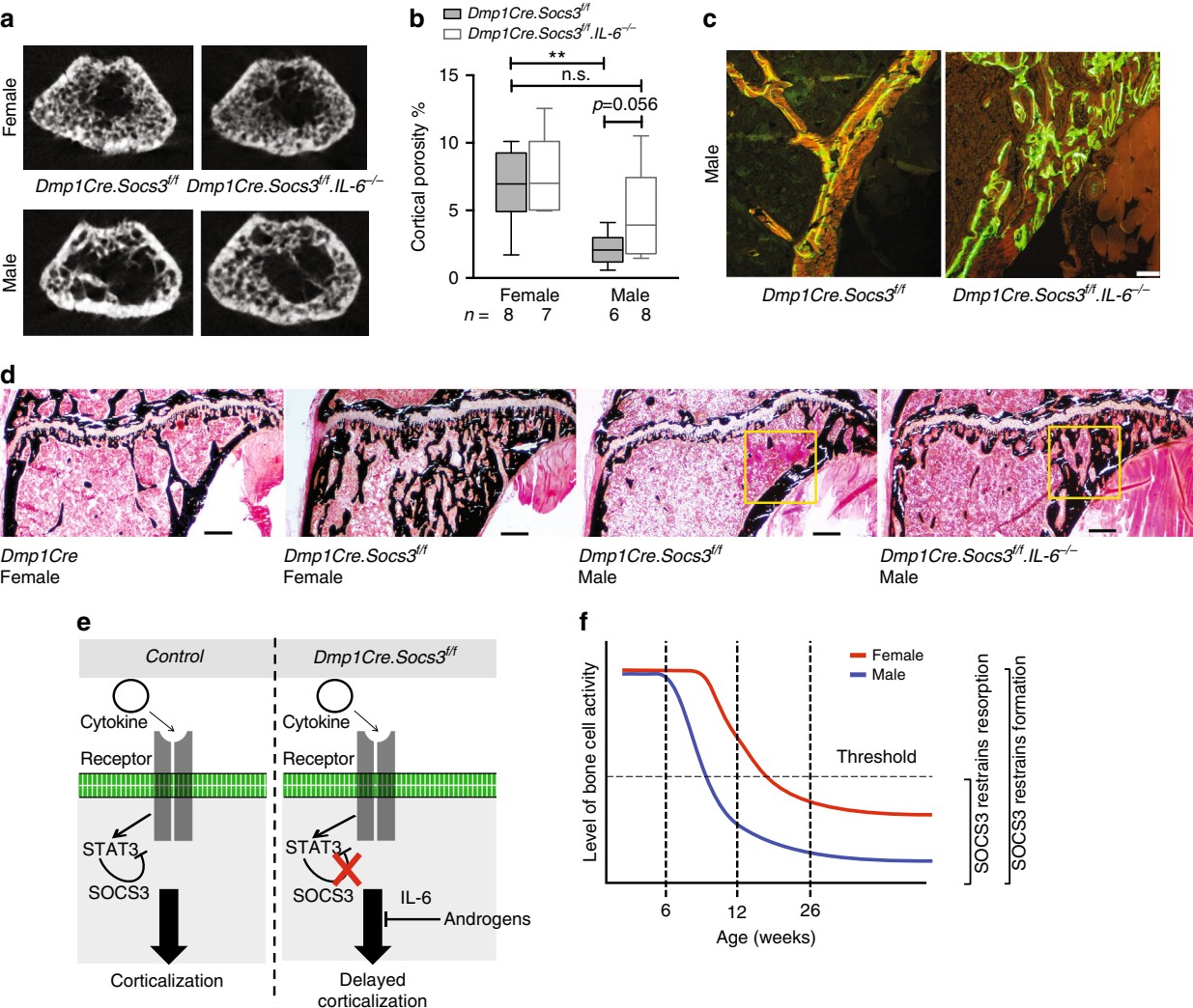

**Fig. 7** IL-6 deletion partially rescues the sex-specific difference in corticalization in *Dmp1Cre.Socs3*^*f/f*^ mice. **a** Representative images of the top slice from the femoral metaphys is of 12-week-old female and male *Dmp1Cre.Socs3*^*f/f*^ and *Dmp1Cre.Socs3*^*f/f*^. *IL-6*^*−/−*^ littermates. **b** Cortical porosity (box plots with whiskers showing minimum and maximum values) in the femoral metaphysis of female and male *Dmp1Cre.Socs3*^*f/f*^ and *Dmp1Cre.Socs3*^*f/f*^. *IL-6*^*−/−*^ littermates; *n* per group is shown below the graph; \*\**p* < 0.01 for comparison shown, determined by two-way ANOVA with Sidak post-hoc test. **c** Calcein labeling in the corticalization zone in male *Dmp1Cre.Socs3*^*f/f*^ and *Dmp1Cre.Socs3*^*f/f*^. *IL-6*^*−/−*^ littermates on the lateral side of the tibial metaphysis (boxed region shown (**d**)). **c** Representative images showing the morphology of longitudinal sections of the lateral side of the tibial metaphysis in control female mice, *Dmp1Cre.Socs3*^*f/f*^ females, *Dmp1Cre.Socs3*^*f/f*^ males, and *Dmp1Cre.Socs3*^*f/f*^.*IL-6*^*−/−*^ males. Scale bar = 100 μm. Note the change in corticalization at the right hand side (lateral edge) of the metaphysis; *yellow* boxes indicate the regions shown (**c**). **e** Schematic diagram illustrating that cytokines acting on receptors on the cell surface of *Dmp1Cre*-expressing cells require inhibition of STAT3 signaling by SOCS3 to promote corticalization. In *Dmp1Cre.Socs3*^*f/f*^ mice, this is lost, leading to delayed corticalization; this delay is overcome by androgens through an interleukin-6 (IL-6)-dependent action. **f** Schematic diagram illustrating the changes in bone cell activity in male (*blue line*) and female (*red line*) mice as they age including the theoretical threshold of remodeling (*horizontal black line*) below which bone resorption is restrained by SOCS3 in *Dmp1Cre* +ve cells

termed endocortical apposition[8, 10]. The extended delay in corticalization in female *Dmp1Cre.Socs3*^*f/f*^ mice compared with males suggests that sex-steroid hormones regulate corticalization through actions on SOCS3-dependent cytokines in *Dmp1Cre*-expressing cells. Sex hormone treatment from the time of gonadectomy confirmed that this was the case. Ovariectomy eliminated the corticalization delay in female mice indicating a requirement for ovarian hormones in that delay. The recapitulation of the phenotype with estradiol treatment in both males and females suggests that estradiol promotes the SOCS3-dependent action that delays corticalization. Alternatively, the rescue of the phenotype with the pure non-aromatizable androgen DHT suggests that androgen action (predominantly mediated by testosterone in vivo) inhibits this effect. As SOCS3 provides negative feedback for a range of cytokines, effects of estradiol or

androgens to modify the expression or action of these cytokines is likely a mechanism by which this occurs. Although our initial studies have not yet identified the hyperactive cytokine/s responsible for the delay in corticalization, genetic deletion of IL-6 showed that pure androgen action promotes corticalization by a mechanism that is partially dependent on IL-6 signaling. The androgen receptor is expressed in both osteoblasts and osteocytes[57], and previous work has shown that DHT inhibits IL-6 secretion induced by treatment of bone marrow-derived stromal cells with inflammatory cytokines[58]. Our result suggests that androgen receptor-mediated regulation of IL-6 in these cells may mediate the male-specific pattern of corticalization.

It has been proposed that although metaphyseal corticalization occurs by trabecular coalescence, cortical bone at the diaphysis develops through sub-periosteal apposition[59]. The phenotype we

observed is specific to the endocortical surface; no changes in periosteal width were detected in male or female *Dmp1Cre.Socs3^{f/f}* mice. Despite this, femoral strength at the diaphysis was lower in 26-week-old female *Dmp1Cre.Socs3^{f/f}* mice compared with controls. This suggests that trabecular coalescence contributes, not only to metaphyseal structure, but it also has a role in determining the strength of the diaphyseal cortex.

What are the mechanisms by which androgen action promotes trabecular coalescence? Our histomorphometric analysis at 12 weeks of age suggests that the lower rate of bone cell activity in male mice contributes to their recovery from delayed corticalization. The normal suppression of both bone formation and resorption at this age in male mice was associated with a recovery of the balance between bone formation and bone resorption in the SOCS3-deficient mice, although both were still elevated. The changing dynamics of the phenotype of *Dmp1Cre.Socs3^{f/f}* mice suggests a model for the way SOCS3 controls bone mass at different stages of life (Fig. 7f). SOCS3 in late osteoblasts and osteocytes appears to restrain bone formation throughout life, by providing negative feedback to those cytokines previously shown to promote bone formation, such as OSM, CT-1, and LIF[16–18]. There appears to be a threshold of bone cell activity below which SOCS3 in osteoblasts and osteocytes also restrains bone resorption, by providing negative feedback to cytokines (such as OSM, CT-1, IL-6, and IL-11) that act on the osteoblast lineage to stimulate production of the pro-osteoclastic factor RANKL[48, 60, 61]. When osteoblast numbers decrease with maturity, which occurs earlier in male mice than in females (compare Figs. 2c and 3c), SOCS3 restrains both bone formation and resorption. In female mice, when their osteoblast numbers become sufficiently low, at 16 weeks of age, their phenotype also begins to recover (Fig. 7f). We have previously reported that signaling via SOCS3 in osteoclast precursors restrains osteoclast formation[62], but in the present study, SOCS3 deletion is restricted to the osteoblast lineage. It is likely that SOCS3 negative feedback is required to limit production of the pro-osteoclastic factor RANKL by osteoblast-lineage cells. Stimulation of osteoclast formation by IL-6 family cytokines such as OSM, CT-1, IL-6, and IL-11[48, 60, 61] requires the presence of osteoblast-lineage cells, who respond to these cytokines by producing RANKL[63, 64]. The lower rate of bone cell activity in adult male mice may be an indirect mechanism by which androgens promote corticalization.

Another possibility is that SOCS3-deficient mice have an elevated set-point at which the mechanical stimulation provided by body weight promotes corticalization. Osteocytes are mechanoresponsive cells[65], and mRNA levels of SOCS3, STAT3, and of SOCS3-dependent cytokines including OSM, leptin, and IL-6 are stimulated by in vivo mechanical loading[66]. As male mice weigh ~4 g more than female mice at 12 weeks of age, and female mice reach that higher weight at 16 weeks of age (Supplementary Fig. 1A), it could be that this is the new weight set-point in *Dmp1Cre.Socs3^{f/f}* mice at which corticalization occurs. Such a model is supported by the effects of DHT treatment, which not only promoted corticalization, but also led to a significant weight gain in both male and female mice (Supplementary Fig. 2A, B). In contrast, estradiol treatment of male mice did not significantly reduce body weight, and neither did IL-6 deletion (Supplementary Fig. 2A–C), so while this is an appealing mechanism, more work is needed to determine whether there is an altered weight-dependent or body composition-dependent mechanical set-point in these mice.

Although corticalization is most active during growth, the delayed trabecular coalescence in *Dmp1Cre.Socs3^{f/f}* mice was not associated with any change in longitudinal growth. This, and the targeting of SOCS3 deletion to non-chondrocytic cells indicates that corticalization is not determined only by events at the epiphyseal growth plate, which is the source of trabeculae in growing bone. This suggests that it is possible to promote trabecular coalescence into cortical bone in adults; once the mechanism has been identified, this could be used therapeutically to promote cortical bone strength, and perhaps to reverse the process of trabecularization that occurs with age and contributes to long-term fracture risk[67]. Although the structure of cortical bone differs between large mammals, which contain osteonal Haversian systems, and small mammals such as rats and mice that have a more simplified cortical structure, both mice and humans form metaphyseal cortical bone by trabecular coalescence. This pathway could also have relevance to the related process by which cortical fracture healing occurs, and SOCS3-dependent pathways could be targeted to promote corticalization at fracture sites, or to protect cortical bone structure in the context of tumor metastasis.

*Dmp1Cre*-directed expression has been detected, not only in osteoblasts and osteocytes, but also in muscle[38], gastrointestinal mesenchymal cells[68], regions of the brain[68], and an unidentified population of bone marrow cells near blood vessels[68]. We do not know whether *Dmp1Cre*-directed recombination of *Socs3* has occurred at these sites in this study, or what effect this may have on corticalization. As muscle-derived factors have been shown to regulate bone mass through a local mechanism[69], we posit that altered STAT3 signaling in the muscle due to *Socs3* deletion would be most likely to alter periosteal growth, but this did not occur in the present study. Nevertheless, we cannot exclude the possibility that *Socs3* cytokine-dependent signaling in other parts of the body may be involved in the process of corticalization.

To summarize, we show that (1) lack of SOCS3 in bones of young mice leads to delayed corticalization, and (2) this is rectified in male mice by androgen action through an IL-6-dependent pathway. We conclude that SOCS3-dependent cytokine signaling in osteocytes has an important role in normal physiology. It determines bone mass during growth, and promotes the coalescence of trabeculae required for the formation of strong cortical bone in adulthood. Furthermore, the sex differences in cortical bone strength that emerge during sexual maturation (puberty) may be determined by effects of androgens on SOCS3-dependent cytokine signaling within the osteocyte network, including an effect on IL-6 signaling.

## Methods

**Mice.** *Dmp1Cre* mice (*Tg(Dmp1-cre)^{1Jqfe}*), backcrossed onto a C57BL/6 background, containing the *Dmp1* 10-kb promoter region, were kindly provided by Lynda Bonewald (formerly University of Kansas, Kansas City, USA)[70]. Floxed *Socs3* mice (*Socs3^{tm1Wsa}*) backcrossed onto C57BL/6 were kindly provided by Warren Alexander (Walter and Eliza Hall Institute of Medical Research, Melbourne, Australia)[23]. To eliminate IL-6 signaling, IL-6 null mice (*Il6^{tm1Kopf}*)[71], backcrossed onto a C57BL/6 background were used. All animal procedures were conducted with approval of the St. Vincent's Health Melbourne Animal Ethics Committee. For all experiments, *Dmp1Cre* littermates or cousins were used as controls; all samples were collected with non-identifiable codes that were allocated sequentially, and therefore were random according to sex, genotype, and treatment; all analyses were performed with observers blinded to the sex, genotype, and treatment of the animals.

To assess the basal phenotype, samples for histomorphometry, micro-CT, mRNA, and serum analyses were collected at 2, 6, 12, 16, and 26 weeks of age; calcein injections were performed at 7 and 3 days prior to tissue collection for 6-week-old mice, at 7 and 2 days prior to tissue collection for 12-week-old mice, and at days 10 and 3 days prior to tissue collection for 26-week-old mice. Sample size for initial phenotypic analysis was selected based on previous studies, with a minimum of 7 mice per group. For the IL-6 null experiment, a minimum of 6 mice were used, based on the severity of the phenotype observed in the first arm of the study. All were fasted overnight prior to tissue collection; blood samples were centrifuged for 10 min at 4000 rpm and serum stored at −80 °C until analysis. When samples from 12-week-old mice were collected, one femur was flushed of marrow and the bone shaft was collected for RNA analyses[22]. Briefly, bones were homogenized with a LS-10-35 Polytron homogenizer in Trizol for $4 \times 5$ s bursts and stored at −80 °C. RNA from each bone was purified using RNeasy lipid tissue minikits (Qiagen), according to manufacturer's instructions. cDNA synthesis from

50 to 100 ng DNase-treated RNA from each was performed with AffinityScript (Agilent Technologies, Santa Clara, CA, USA) per the manufacturer's instructions. Stock cDNA was diluted to a concentration of 5 ng/μl and quantitative real-time PCR was performed on 12.5 ng cDNA in a reaction volume of 10 μl using in-house master mix of 10× AmpliTaq Gold with SYBR Green nucleic acid gel stain (Life Technologies). Primers for Socs3 and B2m are described previously[51, 72].

**Gonadectomy and sex hormone treatments**. To determine the sex hormone dependency of the Dmp1Cre.Socs3[f/f] phenotype, mice were bilaterally sham-operated or gonadectomized at 6 weeks of age. Although under anesthesia, 1 cm silastic implants containing ~10 μg 17-β-estradiol (E$_2$)[73] or ~10 mg dihydrotestosterone (DHT)[74] (both from Steraloids, Newport, RI) were inserted subdermally at the back of the neck. These have been previously shown to allow steroid release at levels previously that partially restore uterine or seminal vesicle weight after gonadectomy[68, 73–75]. Calcein was injected at 7 and 2 days prior to tissue collection at 12 weeks of age. A minimum of 5 mice were used per group, based on the severity of the phenotype observed in the first arm of the study.

**Histomorphometry, serum biochemistry, and micro-CT analysis**. Calcein labels were visualized and histomorphometry was performed using the Osteomeasure system (Osteometrics, Decatur, GA) on undecalcified methylmethacrylate-embedded longitudinal sections of the tibial distal metaphysis, in the secondary spongiosa, commencing 370 μm distal to the base of the growth plate, and extending for 1.11 mm; toluidine blue and Xylenol Orange-stained sections were used[76]. Serum cross-linked C-terminal telopeptide of type I collagen (CTX-1) and N-terminal propeptide of type I collagen (P1NP) levels were determined on samples from 26-week-old mice by ELISA (Immunodiagnostic Systems Limited, Boldon, Tyne & Wear, UK) as per manufacturer's instructions. Ex vivo micro-CT was performed on femoral specimens using a SkyScan 1076 system (Bruker-micro-CT, Kontich, Belgium). Images were acquired using the following settings: 9 μm voxel resolution, 0.5 mm aluminum filter, 50 kV voltage and 100 μA current, exposure time, rotation 0.5°, and frame averaging = 1. Images were reconstructed and analyzed using SkyScan software programs NRecon (version 1.6.8.0), Data-Viewer (version 1.4.4), and CT Analyser (version 1.11.8.0). Regions of interest (RoI) were selected to analyze the secondary spongiosa (trabecular bone) and near to the mid-shaft (cortical bone). For the trabecular RoI, the femoral distal growth plate was identified and measurements were taken in a region commencing at 7.5% of the total femur length towards the femoral mid-shaft and extending for 12.5% of the total femur length in 2-week-old mice, and 15% of the total femur length in 6, 12, 16, and 26-week-old mice. Cortical bone was measured at a site commencing at 25% (in 2-week-old mice) or 30% (for all other ages) of the total femoral length towards the mid-shaft; the RoI was 10% (in 2-week-old mice) or 15% (for all other ages) of the total femur length. Analysis of bone structure was completed using adaptive thresholding (mean of min and max values) in CT Analyser. Thresholds for analysis were determined manually based on grayscale values for each experimental group as follows: trabecular bone: 2-week-old mice: 68–255; all other ages: 45–255; cortical bone: 100–255 at all ages. To generate cross-sectional thresholded images, Paraview 13.4.1 was used with 200 iterations. For the gonadectomy study, bones were scanned and analyzed as above with the modification that images were acquired at 45 kV and 220 μA using NRecon (version 1.6.10.2) and CT Analyser (version 1.15.4.0). For trabecular analysis in the 5th lumbar vertebrae, an RoI of half the height of the bone (vertically centered) with a diameter 2/3 the width of the vertebral body was used.

**Mechanical loading**. The mechanical properties of isolated tibiae from 26-week-old mice were assessed by 3 point-bending tests with a constant span length of 8 mm on a Bose Biodynamic 5500 Test Instrument (Bose, DE, USA). Before testing, tibiae were kept moist in gauze swabs soaked in phosphate-buffered saline (PBS). The bone was positioned horizontally with the anterior surface upwards, centered on the supports, and the pressing force was directed vertically to the mid-shaft of the bone; two bones showed abnormal force-displacement curves, likely due to rolling during the test, and were excluded. Each bone was compressed at a constant 0.5 mm/s until failure. WinTest software was used to collect the load-displacement data at 250 data points per second for a total of 10 s. Structural properties including Ultimate force (F$_U$; N), yield force (F$_Y$; N), stiffness (S; N/mm), and energy (work) to failure (U; J) endured by the tibiae were calculated from load/displacement data[77]. The yield point was determined from the load deformation curve at the point at which the curve deviated from linear. Widths of the cortical mid-shaft in the mediolateral (ML) and anteroposterior (AP) directions, moment of inertia, and average cortical thickness were determined by micro-CT; this was combined with 3 point-bending data to calculate material-specific properties[78].

**Statistical analysis**. Significant differences were identified by two-way ANOVA with Sidak post-hoc test, as indicated in each figure legend. $p < 0.05$ was considered significant.

**Data availability**. All data are available from the authors upon reasonable request.

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

## Acknowledgements

This work was funded by a National Health and Medical Research (NHMRC) Project Grant 1058625 to N.A.S. and T.J.M. N.A.S. is supported by an NHMRC Senior Research Fellowship. The Victorian State Government Operational Infrastructure Support Scheme provides support to St. Vincent's Institute. We thank BRC and EMSU staff for excellent animal care, Joshua Johnson for histology technical assistance, and Mark Jimenez for preparing silastic implants for this study.

## Author contributions

N.A.S. designed the study, interpreted the data, and wrote the manuscript. D.-C.C., H.J.B., R.W.J., J.H.G., B.A.T., I.J.P., E.C.W. and N.E.M. conducted experiments, carried out laboratory-based analyses, analyzed data, prepared figures, and contributed to the interpretation of the data. D.J.H. provided valuable advice on the design of the gona-dectomy experiment and provided silastic implants. T.J.M. provided advice on experimental design and interpretation of data. All authors revised the manuscript and approved the final version.

## Additional information

**Competing interests:** The authors declare no competing financial interests.

