## [Peer Review file · Nature Communications]

Reviewers' comments:

Reviewer #1 (Remarks to the Author):

The manuscript entitled "Androgen action promotes bone corticalization by inhibiting SOCS3-dependent interleukin-6 signalling" by Sims and colleagues details a thorough investigation into bone growth abnormalities that arise from deletion of SOCS3 in osteocytes.

The authors find that SOCS3 deletion induces high trabecular bone volume (and therefore a defect in bone strength) in both male and female pre-pubescent mice but that in male mice this phenotype is resolved during adulthood. Addition of testosterone largely protects against this phenotype whilst estrogen exacerbates it. SOCS3 is a strong inhibitor of IL-6 family cytokines and these studies suggest a role for those cytokines in bone growth. This is in keeping with previous studies showing a role for gp130-family cytokines in bone growth and homeostasis and the authors are experts in this field.

As I have no expertise in bone biology I will be restricting my comments to other aspects of the study.

My only major issue is my confusion regarding the link the authors make between the role of androgen (specifically testosterone) and IL-6. My understanding of the data presented in Figure 7 is that removing IL-6 from the SOCS KO mice has little if any effect on female mice but exacerbates the phenotype (cortical porosity) in male mice. This leads the authors to conclude that the delayed corticalization (induced by loss of SOCS3) is "inhibited by androgens through interleukin 6 (IL-6)" (Figure legend, Figure 7).

If I understand that sentence correctly the authors are stating that androgen REQUIRES IL-6 signalling to promote corticalization. This suggests that removal of IL-6 from the system will reverse the androgen-dependent effects they observe, which appears to be the case.

However in several places in the manuscript the authors appear to state that androgen INHIBITS IL-6 signalling (in order to reverse the delayed corticalization effects). In this case one would expect that removing IL-6 signalling in female mice would exacerbate the phenotype but that in male mice (where androgen is already inhibiting IL-6) it would make less of a difference, which appears to be the opposite of what is shown in Figure 7.

There are several sentences throughout the manuscript that imply that androgen INHIBITS IL-6 signalling:

Results "This suggests that the androgen-dependent rescue of delayed corticalization in male Dmp1Cre.Socs3f/f mice is caused, at least in part, by suppression of IL-6 signaling."

Abstract "suggesting that androgen action promotes metaphyseal corticalization by inhibiting IL-6 signaling"

Introduction "sex-differences in corticalization may be explained, at least in part, by inhibition of local IL-6 signaling by androgens"

Title "Androgen action promotes bone corticalization by inhibiting SOCS3-dependent interleukin-6 signalling"

Those above sentences appear to contrast the following sentences:

Figure 7 legend: "In Dmp1Cre.Socs3f/f mice, this inhibition is lost, leading to delayed corticalization, which is inhibited by androgens through interleukin 6 (IL-6)"

Start of discussion "pure androgen action (mainly mediated by testosterone in vivo) promotes corticalization through an IL-6 dependent pathway"

End of discussion "pure androgen action promotes corticalization by a mechanism that is partially dependent on IL-6 signaling"

My confusion as outlined above may very well be a lack of understanding on my part and I would welcome clarification by the authors

Minor comments:

Is testing the efficacy of SOCS3 deletion by RT-PCR as shown in Figure 1A really the best method? As this shows only a slight reduction in SOCS3 mRNA levels.

Figure 1b: The dashed lines are not clear in the figure legend.

Page 10: The sentence "Thickened" cortical bone was measured throughout the metaphyseal region and partitioned into the spongy, incompletely coalesced cortical component versus normal compact cortical bone, we observed that the incompletely corticalized (spongy) bone" does not appear to make sense

Tb.Sp is not defined

Abstract, line 48 "mic" should read "mice"

It would be interesting to see if loss of gp130 reverses the phenotype of the SOCS3 KO osteocytes

Reviewer #2 (Remarks to the Author):

The authors of the manuscript study the effect of knockout of SOCS3 specifically in osteocytes using the DMP1cre. The authors find a very interesting phenotype where there is initially an increase in the trabecular bone phenotype with a delay in corticalization in females that persists. Treating females with androgen rescues the cortical phenotype and treating males with estradiol recapitulates the corticalization delay in males. Suggesting that testosterone through IL6 mediates corticalization. The bone phenotype is thoroughly and extensively characterized. The authors use OVX and ORX to thoroughly explore the sex dependent phenotypes. The signaling mechanism is partially shown to be through IL6 by using null mice. Though most of the data is descriptive it is very well written with the study being very novel and general interest to the bone field. I would support publication with the below mentioned revisions.

The major drawback is the lack of molecular data showing the mechanism. Having some in vitro data to support the mechanism would be very beneficial and would significantly add to this solid data set.

Is there any molecular data showing the involvement of STAT signaling?

Some additional information about the DMP1cre would be helpful with the new data showing that the 14kb DMP1cre has a promiscuous expression pattern it would be useful if the authors could comment on the specificity of the cre line they utilized.

Reviewer #3 (Remarks to the Author):

This is an interesting and carefully performed study defining the development of metaphyseal cortical bone and the possible role of sex steroids. It shows an important role for the family of cytokines regulated by SOCS3 in this process, and it is possible that targeting this pathway could enhance metaphyseal cortical bone in adults and therefore be of therapeutic importance.

The paper can be difficult to follow in places, as it uses complex histology and imaging. Nonetheless, the data are presented well overall. My main concern pertains to the IL6 story. As I understand this, deleting SOCS3 leads to increased activity of a family of cytokines, including IL6. In females, deleting IL6 does not rescue the phenotype, so IL6 is not the cause. In males, deleting IL6 moves them towards a female phenotype (Fig 7B) – and yet the authors conclude that DHT suppression of IL6 is part of the reason for the sexual divergence. The main part that does not make sense is how this explanation fits with the data that deleting IL6 in males makes cortical porosity look more like females? If DHT is suppressing IL6, and IL6 is part of the reason for the delayed corticalization, then deleting IL6 in males should have either no effect (IL6 already suppressed) or actually make the males look even more different from the females (IL6 even lower)? It seems I am missing something here...

A second point related to cortical bone in the metaphysis vs diaphysis. As shown by Cadet et al. (JBJS 85A: 1739, 2003), cortical bone at the metaphysis does develop by trabecular condensation. However, diaphyseal cortical bone develops through sub-periosteal apposition. And yet in Fig 6, the authors present very similar findings at both the metaphysis and diaphysis. Do they have an explanation for this?

Other, minor points:

- Were all mice in the C57BL/6 background?
- Given that E2 and perhaps DHT often lead to metaphyseal sclerosis when given by pellet or pump, was this a confounder in the study? How “physiological” were these doses? What about uterine or seminal vesicle weights as biomarkers for replacement?
- Fig 1A – the extent of deletion of SOCS3 is surprisingly small. Generally the DMPCre gives better results in terms of reduction of the specific mRNA. Any reasons for what appears to be relatively poor deletion?
- Fig 7B – to argue that the IL6 KO males were no longer significantly different from the females may be true in terms of the p value, but they do still look quite different, and may well be with a somewhat larger sample size. So the authors do need to be more cautious regarding this contention.

Reviewers' comments and responses:

Reviewer #1:

COMMENT: My only major issue is my confusion regarding the link the authors make between the role of androgen (specifically testosterone) and IL-6. My understanding of the data presented in Figure 7 is that removing IL-6 from the SOCS KO mice has little if any effect on female mice but exacerbates the phenotype (cortical porosity) in male mice. This leads the authors to conclude that the delayed corticalization (induced by loss of SOCS3) is “inhibited by androgens through interleukin 6 (IL-6)” (Figure legend, Figure 7). If I understand that sentence correctly the authors are stating that androgen REQUIRES IL-6 signalling to promote corticalization. This suggests that removal of IL-6 from the system will reverse the androgen-dependent effects they observe, which appears to be the case. However in several places in the manuscript the authors appear to state that androgen INHIBITS IL-6 signalling (in order to reverse the delayed corticalization effects). In this case one would expect that removing IL-6 signalling in female mice would exacerbate the phenotype but that in male mice (where androgen is already inhibiting IL-6) it would make less of a difference, which appears to be the opposite of what is shown in Figure 7.

There are several sentences throughout the manuscript that imply that androgen INHIBITS IL-6 signalling:

Results “This suggests that the androgen-dependent rescue of delayed corticalization in male *Dmp1Cre.Socs3f/f* mice is caused, at least in part, by suppression of IL-6 signaling.”

Abstract “suggesting that androgen action promotes metaphyseal corticalization by inhibiting IL-6 signaling”

Introduction “sex-differences in corticalization may be explained, at least in part, by inhibition of local IL-6 signaling by androgens”

Title “Androgen action promotes bone corticalization by inhibiting SOCS3-dependent interleukin-6 signalling”

Those above sentences appear to contrast the following sentences:

Figure 7 legend: “In *Dmp1Cre.Socs3f/f* mice, this inhibition is lost, leading to delayed corticalization, which is inhibited by androgens through interleukin 6 (IL-6)”

Start of discussion “pure androgen action (mainly mediated by testosterone in vivo) promotes corticalization through an IL-6 dependent pathway”

End of discussion “pure androgen action promotes corticalization by a mechanism that is partially dependent on IL-6 signaling”

My confusion as outlined above may very well be a lack of understanding on my part and I would welcome clarification by the authors.

RESPONSE: The reviewer has helped us to think about our data in a new way. As the reviewer points out, since deletion of IL-6 in females had no effect on the phenotype, it is more likely that the effect of androgens on corticalisation is due to a stimulation of IL-6 rather than an inhibition. In the original manuscript we showed that androgens block (inhibit) the delay in corticalisation in *Dmp1Cre.Socs3^{ff}*

mice through an IL-6-dependent action. We had hypothesized that was caused by inhibition of IL-6 based on a previous report that DHT inhibited IL-6 secretion induced in a stromal cell line by IL-1/TNF treatment (Bellido 1995). However, this may not relate directly to our study, in which there is no inflammation. We have removed the speculation about IL-6 being inhibited by DHT, and changed the manuscript to state simply that the ability of androgens to promote corticalisation is dependent on IL-6.

We also realise from the reviewer's comments that the distinction between the inhibition of delayed corticalization, and the inhibition of IL-6 signalling was unclear. We have rephrased some of the comments above to clarify our meaning.

The following lines have been modified in response to this comment: 43, 93, 258-259, Title, Figure 7 legend.

Minor comments:

COMMENT: Is testing the efficacy of SOCS3 deletion by RT-PCR as shown in Figure 1A really the best method? As this shows only a slight reduction in SOCS3 mRNA levels.

RESPONSE: Since SOCS3 is expressed ubiquitously, although we flushed marrow out of the bone samples, it is likely that this was only partially effective at removing non-osteocytic cells, and the *Socs3* mRNA we detect is from these contaminating cells. Similar reductions (<50%) have been observed in other mice where ubiquitous, or near ubiquitous, genes have been targeted by *Dmp1Cre* (Johnson et al JBMR 2014 – targeted deletion of *Il6st* / gp130; Fulzele et al, Blood 2013 – targeted deletion of *Gsa* / Gs alpha; Windahl et al, PNAS 2013 – targeted deletion of *Esr1* / Estrogen receptor alpha).

Unfortunately, we have not been able to find an antibody for SOCS3 that is effective in bone samples to confirm SOCS3 deletion at a cell-specific level. **We have added a comment about cellular purity of the RNA samples in the results section, including reference to the previously published manuscripts showing *Dmp1Cre*-directed gene deletion of other ubiquitous genes (lines 102-105)**

COMMENT: Figure 1b: The dashed lines are not clear in the figure legend.

RESPONSE: The legend has been corrected to make the dashed lines clear.

COMMENT: Page 10: The sentence “Thickened” cortical bone was measured throughout the metaphyseal region and partitioned into the spongy, incompletely coalesced cortical component versus normal compact cortical bone, we observed that the incompletely corticalized (spongy) bone” does not appear to make sense

RESPONSE: This sentence has been corrected (**line 208**).

COMMENT: Tb.Sp is not defined

RESPONSE: This has been corrected (**line 120**).

COMMENT: Abstract, line 48 “mic” should read “mice”

RESPONSE: This sentence was removed when shortening the abstract to the required length.

COMMENT: It would be interesting to see if loss of gp130 reverses the phenotype of the SOCS3 KO osteocytes

RESPONSE: We agree, and are currently making the genetic crosses required to answer this question, but it is beyond the scope of the current paper.

Reviewer #2

COMMENT: The authors of the manuscript study the effect of knockout of SOCS3 specifically in osteocytes using the *DMP1cre*. The authors find a very interesting phenotype where there is initially an

increase in the trabecular bone phenotype with a delay in corticalization in females that persists. Treating females with androgen rescues the cortical phenotype and treating males with estradiol recapitulates the corticalization delay in males. Suggesting that testosterone through IL6 mediates corticalization. The bone phenotype is thoroughly and extensively characterized. The authors use OVX and ORX to thoroughly explore the sex dependent phenotypes. The signaling mechanism is partially shown to be through IL6 by using null mice. Though most of the data is descriptive it is very well written with the study being very novel and general interest to the bone field. I would support publication with the below mentioned revisions.

The major drawback is the lack of molecular data showing the mechanism. Having some *in vitro* data to support the mechanism would be very beneficial and would significantly add to this solid data set. Is there any molecular data showing the involvement of STAT signaling?

RESPONSE: We agree with the reviewer that molecular data would be helpful, but the effect of SOCS3 cytokines on corticalisation are (1) site-specific (only in metaphyseal bone), (2) age-specific (varies with maturity and stages of bone development), (3) modified by systemic sex-steroids, and (4) likely to be modified by mechanical loading. For these reasons, we do not think it is possible to use cultured cells to identify the mechanism for this phenotype; our continuing work is focussing on using *in vivo* approaches.

COMMENT: Some additional information about the DMP1cre would be helpful with the new data showing that the 14kb DMP1cre has a promiscuous expression pattern it would be useful if the authors could comment on the specificity of the cre line they utilized.

RESPONSE: We have added a short paragraph on the potential limits of this study due to non-specific Dmp1Cre expression (**Lines 418-427**)

Reviewer #3

COMMENT: The paper can be difficult to follow in places, as it uses complex histology and imaging. Nonetheless, the data are presented well overall. My main concern pertains to the IL6 story. As I understand this, deleting SOCS3 leads to increased activity of a family of cytokines, including IL6. In females, deleting IL6 does not rescue the phenotype, so IL6 is not the cause. In males, deleting IL6 moves them towards a female phenotype (Fig 7B) – and yet the authors conclude that DHT suppression of IL6 is part of the reason for the sexual divergence. The main part that does not make sense is how this explanation fits with the data that deleting IL6 in males makes cortical porosity look more like females? If DHT is suppressing IL6, and IL6 is part of the reason for the delayed corticalization, then deleting IL6 in males should have either no effect (IL6 already suppressed) or actually make the males look even more different from the females (IL6 even lower)? It seems I am missing something here...

RESPONSE: Please see response to Reviewer 1's major comment.

COMMENT: A second point related to cortical bone in the metaphysis vs diaphysis. As shown by Cadet et al. (JBJS 85A: 1739, 2003), cortical bone at the metaphysis does develop by trabecular condensation. However, diaphyseal cortical bone develops through sub-periosteal apposition. And yet in Fig 6, the authors present very similar findings at both the metaphysis and diaphysis. Do they have an explanation for this?

RESPONSE: We observed no difference in periosteal width at the diaphysis, suggesting that sub-periosteal apposition is normal in our mouse model. We have added this data to Supplementary Figure 1, and describe it in lines **109-110**. Even though periosteal growth was unaffected, femoral strength at the diaphysis was modified in older female mice. This suggests that although sub-periosteal apposition plays a role in diaphyseal cortical development, as suggested by Cadet, it may not be the exclusive

mediator of diaphyseal development, and trabecular coalescence may also be involved. We have added a note about this in the discussion (lines **350-357**).

In Figure 6, the region that we termed “diaphysis” is not the central diaphysis, but a region that is closer to the diaphysis than the metaphyseal region. To avoid confusion, we have changed the terminology used in Figure 6 to “upper metaphysis” and “lower metaphysis” to avoid confusion (line **230**, and **Figure 6**).

Other, minor points:

COMMENT: Were all mice in the C57BL/6 background?

RESPONSE: Yes, this detail has been added to the Methods section (lines **439 and 444**).

COMMENT: Given that E2 and perhaps DHT often lead to metaphyseal sclerosis when given by pellet or pump, was this a confounder in the study? How “physiological” were these doses? What about uterine or seminal vesicle weights as biomarkers for replacement?

RESPONSE: There was no metaphyseal sclerosis associated with the doses of E₂ and DHT used in this study. We used doses previously shown to retain uterine and seminal vesicle weights after gonadectomy. We did not measure uterine or seminal vesicle weights in this experiment. We quantified the effects of E₂ and DHT on trabecular bone at the vertebrae because an effect on corticalisation would not impact on the phenotype at this site (Figure 5) and did not observe sclerosis at that site. We also did not observe metaphyseal sclerosis due to E₂ or DHT in femoral samples, as indicated by the metaphyseal images shown in Figure 6A, C. We have added a note to clarify doses used in the methods, including specific references to those papers that measured the effects of these implants on uterine and seminal vesicle weights (lines **478-480**) and have added a note that there was no bone sclerosis in the Results section (lines **191-193**).

COMMENT: Fig 1A – the extent of deletion of SOCS3 is surprisingly small. Generally the DMPCre gives better results in terms of reduction of the specific mRNA. Any reasons for what appears to be relatively poor deletion?

RESPONSE: Please see our response to Reviewer 1’s first minor comment. With ubiquitously expressed genes, the DMP1Cre is generally only about 50% effective at reducing target gene levels in whole mRNA samples of marrow-flushed bones, likely due to contaminating cells.

COMMENT: Fig 7B – to argue that the IL6 KO males were no longer significantly different from the females may be true in terms of the p value, but they do still look quite different, and may well be with a somewhat larger sample size. So the authors do need to be more cautious regarding this contention.

RESPONSE: This was already noted at the end of the Introduction and Results sections, and in both the first paragraph and latter part of the Discussion, where we said that this was partially rescued, indicating “at least partial” dependency on IL-6. We have now modified the last line of the discussion to further soften this assertion (lines **433-435**).

REVIEWERS' COMMENTS:**Reviewer #1 (Remarks to the Author):**

All of my queries have been answered. My only final comment is that the manuscript would benefit from a single sentence that summarises both parts of the model that the authors suggest-by that I mean that (1) lack of SOCS3 in young mice (and hence presumably overactive IL-6 family signalling) leads to poor bone formation but (2) that this is rectified in older male mice by androgen acting through IL-6.

Reviewer #2 (Remarks to the Author):

The authors have addressed my original comments.

Reviewer #3 (Remarks to the Author):

The authors have satisfactorily addressed my concerns.

MS ID#: NCOMMS-17-04377B

MS TITLE: Bone corticalization requires local SOCS3 activity and is promoted by androgen action via interleukin-6

Reviewers' comments and responses:

Reviewer #1:

COMMENT: All of my queries have been answered. My only final comment is that the manuscript would benefit from a single sentence that summarises both parts of the model that the authors suggest- by that I mean that (1) lack of SOCS3 in young mice (and hence presumably overactive IL-6 family signalling) leads to poor bone formation but (2) that this is rectified in older male mice by androgen acting through IL-6.

RESPONSE: We have added this comment as a summary in the final paragraph of the Discussion (lines 445-447).

Reviewer #2:

COMMENT: The authors have addressed my original comments.

Reviewer #3:

COMMENT: The authors have satisfactorily addressed my concerns.